# Prediction of Greenhouse Tomato Crop Evapotranspiration Using XGBoost Machine Learning Model

**DOI:** 10.3390/plants11151923

**Published:** 2022-07-25

**Authors:** Jiankun Ge, Linfeng Zhao, Zihui Yu, Huanhuan Liu, Lei Zhang, Xuewen Gong, Huaiwei Sun

**Affiliations:** 1School of Water Conservancy, North China University of Water Resources and Electric Power, Zhengzhou 450000, China; gejiankun@ncwu.edu.cn (J.G.); zhaolinfeng0928@163.com (L.Z.); yuzihui1006@163.com (Z.Y.); liu2160214044@163.com (H.L.); zhanglei@ncwu.edu.cn (L.Z.); 2School of Civil and Hydraulic Engineering, Huazhong University of Science and Technology, Wuhan 430030, China; hsun@hust.edu.cn

**Keywords:** XGBoost regression, evapotranspiration, solar greenhouse, drip irrigated tomato, machine learning

## Abstract

Crop evapotranspiration estimation is a key parameter for achieving functional irrigation systems. However, ET is difficult to directly measure, so an ideal solution was to develop a simulation model to obtain ET. There are many ways to calculate ET, most of which use models based on the Penman–Monteith equation, but they are often inaccurate when applied to greenhouse crop evapotranspiration. The use of machine learning models to predict ET has gradually increased, but research into their application for greenhouse crops is relatively rare. We used experimental data for three years (2019–2021) to model the effects on ET of eight meteorological factors (net solar radiation (*R_n_*), mean temperature (*T_a_*), minimum temperature (*T_amin_*), maximum temperature (*T_amax_*), relative humidity (RH), minimum relative humidity (RH_min_), maximum relative humidity (RH_max_), and wind speed (V)) using a greenhouse drip irrigated tomato crop ET prediction model (XGBR-ET) that was based on XGBoost regression (XGBR). The model was compared with seven other common regression models (linear regression (LR), support vector regression (SVR), K neighbors regression (KNR), random forest regression (RFR), AdaBoost regression (ABR), bagging regression (BR), and gradient boosting regression (GBR)). The results showed that *R_n_*, *T_a_*, and *T_amax_* were positively correlated with ET, and that *T_amin_*, RH, RH_min_, RH_max_, and V were negatively correlated with ET. *R_n_* had the greatest correlation with ET (r = 0.89), and V had the least correlation with ET (r = 0.43). The eight models were ordered, in terms of prediction accuracy, XGBR-ET > GBR-ET > SVR-ET > ABR-ET > BR-ET > LR-ET > KNR-ET > RFR-ET. The statistical indicators mean square error (0.032), root mean square error (0.163), mean absolute error (0.132), mean absolute percentage error (4.47%), and coefficient of determination (0.981) of XGBR-ET showed that XGBR-ET modeled daily ET for greenhouse tomatoes well. The parameters of the XGBR-ET model were ablated to show that the order of importance of meteorological factors on XGBR-ET was *R_n_* > RH > RH_min_> *T_amax_*> RH_max_> *T_amin_*> *T_a_*> V. Selecting *R_n_*, RH, RH_min_, *T_amax_*, and *T_amin_* as model input variables using XGBR ensured the prediction accuracy of the model (mean square error 0.047). This study has value as a reference for the simplification of the calculation of evapotranspiration for drip irrigated greenhouse tomato crops using a novel application of machine learning as a basis for an effective irrigation program.

## 1. Introduction

The tomato is a kind of delicious vegetable, which is indispensable on the dinner table of our people, and modern consumers seek to improve both the appearance and nutritional value of it. Therefore, in the context of water scarcity, it is necessary to study the physiological ecology of the tomato based on modern agricultural production techniques to improve its yield and quality. Devising an accurate mathematical crop evapotranspiration (ET) model is fundamental to improving crop water use efficiency and thus important in developing a practical irrigation system for a greenhouse [1,2]. Evapotranspiration is an important parameter in the study of dynamic change in a field water cycle [3] and energy balance [4] and is therefore of great concern in agricultural production.

ET of greenhouse crops is governed by many factors. Most approaches to developing an effective irrigation program depend on the creation of an accurate mathematical model of evapotranspiration [5,6,7]. Current models that predict ET for greenhouse crops consist mainly of empirically based equations using modern mathematical algorithms. The empirical equations draw on energy balance and water vapor diffusion theories to create predictive models of ET that are primarily driven by meteorological factors; such models are robust and widely applied. Commonly used models include the single source Penman–Monteith (P-M) model [8], the dual source Shuttleworth–Wallace (S-W) model [9], the energy-based Priestley–Taylor model [10], and the dual crop coefficient model [11]. These models have been applied in the study of tomato evapotranspiration [12]. However, these models are flawed: the P-M model must improve its use of resistance parameters; the S-W model has many parameters and is unduly influenced by several resistance parameters; the empirical coefficients of the P-T model need to be corrected for different locations; and the dual crop coefficient method lacks consideration of ground cover.

As computer technology has advanced in recent years, many researchers have started to use modern mathematical algorithmic techniques, such as machine learning, in modeling crop ET. For example, Li et al. (2020) [13] developed a model to predict plant transpiration that used random forest regression with plant and environmental parameters. Ahmed et al. (2022) [14] modeled reference crop evapotranspiration using an artificial neural network combined with limited meteorological information. Jiang and Chen (2018) [15] addressed the drawbacks of BP neural networks to create a GA-BP neural network based on the crop water demand prediction model, thus increasing the applicability of ANN models. Darouich et al. (2021) [16] created the SIMDualKc model which predicted daily reference jute water requirements, and this research has provided a body of baseline references for developing models that predict greenhouse crop ET, but there are still obstacles to overcome in developing such a model. For example, support vector regression is slow for large scale training datasets, and it is sensitive to missing data, parameters, and kernel functions; however, a neural network requires a large quantity of data. The weak classifiers that AdaBoost relies on tend to take a long time to train. Random forest algorithms and bagging algorithms may underfit.

In light of the advantages and drawbacks just described, we decided to use the popular technique of extreme gradient boosting to improve ET prediction for greenhouse crops. Extreme gradient boosting (XGBoost) is currently the fastest and best open source boosted tree toolkit [17,18,19] and the algorithm is an improved gradient boosting decision tree (gbdt) that can be used for both classification and regression applications [20]. It is widely used in various fields in applications such as predicting gaseous emission pollutants from buses [21], predicting gene expression [22], and medical patient classification [23]. XGBR (extreme gradient boosting regression) is highly effective in modeling small- and medium-sized datasets; it is flexible and very scalable [24]. XGBR incorporates multiple tree models to build a stronger learning model, which is advantageous for ET prediction as ET is governed by many factors. It is clear that XGBR has great value in predicting evapotranspiration of crops.

In machine learning, it is often necessary to introduce additional factor parameters to obtain accurate model results, which increases the difficulty of observational measurement. How to reduce the numbers of factors and parameters without decreasing model accuracy is therefore a challenge for model application. Ablation experiments can filter out redundant parameters in validating a model [25,26]. This technique identifies the key characteristic variables of the model by removing some modules of the model to see whether model performance is changed, and it can reduce model overfitting caused by excessive numbers of parameters. Ablation has been successfully used in intrusion detection systems [27], driver fatigue detection [28], and crop recognition [29]. We chose many meteorological factors to fit ET in order to increase model accuracy; whether or not the number of model parameters can be reduced using ablation needed to be demonstrated. The objectives of this study were as follows: the input variables were determined through distributions of meteorological factors and correlation coefficients, and eight regression algorithms, including XGBR, were used to model drip irrigated greenhouse tomato ET and analyze the model prediction results. In the process of model optimization, the number of input meteorological factors were gradually reduced using ablation experiments to construct a more reasonable model for predicting drip irrigated greenhouse tomato ET.

## 2. Research Results

### 2.1. Analysis of the Normal Distribution Patterns of ET and Meteorological Factors

The selected data were compared with a normal distribution and other standard distributions to determine the type of distribution of the selected meteorological and ET data. Figure 1 shows the distributions of evapotranspiration (ET), net solar radiation (*R_n_*), temperature (*T_a_*), minimum temperature (*T_amin_*), maximum temperature (*T_amax_*), relative humidity (RH), minimum relative humidity (RH_min_), maximum relative humidity (RH_max_), and wind speed (V). The blue curve in Figure 1 shows the fitting of the normal distribution for each factor, and the black curve is the fitted standard distribution curve. From Figure 1, we can see that the fitted curves of ET, *T_a_*, *T_amax_*, and RH_min_ were similar to the standard normal distribution curve, with similar skewness and kurtosis. Kurtosis of the fitted curves for *R_n_*, *T_amin_*, RH, RH_max_, and V differed from the standard normal distribution curve, which indicated that the distributions of the ET, *T_a_*, *T_amax_*, and RH_min_ data were approximately normal and that the distributions of the *R_n_*, *T_amin_*, RH, RH_max_, and V data need to be further analyzed.

The Shapiro–Wilk test was also used to determine if the sample data were normally distributed. A normal probability plot was used to fit the probability distribution of the data in conjunction with the Shapiro–Wilk test [30]. Figure 2 shows the probability distributions of ET and the eight meteorological factors (Figure 2) with the normal probability plot. The Shapiro–Wilk test produced significant values (*p* > 0.05) for ET and the eight meteorological factors, respectively: 0.318, 0.132, 0.087, 0.267, 0.198, 0.303, 0.073, 0.061, and 0.053. This analysis shows that ET and the meteorological data selected for this study were approximately normally distributed and can therefore be used in the model to predict changes in ET.

### 2.2. Correlations of ET and Meteorological Factors

Analysis of correlations between different variables was undertaken before the model was created. Correlations between the dependent variable ET and the eight independent variables (*R_n_*, *T_a_*, *T_amin_*, *T_amax_*, RH, RH_min_, RH_max_, V) are shown in Figure 3. It can be seen that *R_n_*, *T_a_*, and *T_amax_* were positively correlated with ET and that *T_amin_*, RH, RH_min_, RH_max_, and V were negatively correlated with ET. *R_n_* had the greatest correlation with ET (R = 0.89) and V was least correlated with ET (R = 0.43). It can also be seen that some meteorological factors were correlated. We therefore decided to use all eight meteorological variables as independent variables when creating the prediction model.

### 2.3. Analysis of Model Accuracy

#### 2.3.1. XGBR-ET Model Training and Testing

Figure 4 shows variation in error during model training using XGBR with eight meteorological factors as the independent variables and ET as the dependent variable. Training error on both the training and testing datasets gradually decreased as model training increased to 500 terminating iterations of XGBR. At this point, the error value of the training set was 0.002 and the error value of the test set was 0.048, a difference of 0.046. It can be seen that the fitting of XGBR was good for both the training and test datasets.

#### 2.3.2. Analysis of the Predictions of Different Models

We selected seven regression algorithms to compare with XGBoost regression: traditional linear regression (LR), support vector regression (SVR), K neighbors regression (KNR), random forest regression (RFR), and the computer regression algorithms AdaBoost regression (ABR), bagging regression (BR), and gradient boosting regression (GBR). We developed prediction models for greenhouse tomato crop ET based on the eight regression algorithms using the eight meteorological factors (*R_n_*, *T_a_*, *T_amin_*, *T_amax_*, RH, RH_min_, RH_max_, and V) as input variables and ET as the output variable. The prediction results of the eight models are shown in Figure 5.

The output ET values of the eight models were normalized to permit comparison of accuracy between them. Figure 5 shows that BR-ET, XGBR-ET, and GBR-ET performed better than other models and KNR-ET, RFR-ET, and ABR-ET performed relatively badly. MSE values obtained from the training and testing of the eight models can be compared in Figure 6.

The test set results show that the best performing of the eight models was XGBR-ET and the worst performing was KNR-ET. MSE for ABR-ET and RFR-ET was also >0.05, which indicates that ABR-ET and RFR-ET were not sufficiently accurate in predicting ET. MSE values of the three linear models (LR-ET, SVR-ET, and KNR-ET) were ranked in decreasing order KNR-ET > LR-ET > SVR-ET; the greatest prediction accuracy was given by SVR-ET. In the overall analysis, XGBR-ET produced the most accurate predictions of ET.

MSE, RMSE, MAE, MAPE, and *R*^2^ for the eight models allowed us to evaluate the models. Table 1 shows that MSE and RMSE were identically ordered for the eight models: in descending order, KNR-ET > RFR-ET > ABR-ET > LR-ET > SVR-ET > BR-ET > GBR-ET > XGBR-ET. MSE for XGBR-ET was 0.032 and RMSE was 0.163; these values were, respectively, 21.88% and 25.77% less than those for GBR-ET, which was second to XGBR-ET in prediction accuracy, and, respectively, 259.38% and 85.89% less than those for KNR-ET, which was the least accurate model. MAE and MAPE for the eight models were ordered KNR-ET > ABR-ET > RFR-ET > LR-ET > SVR-ET > BR-ET > GBR-ET > XGBR-ET. This ordering was similar to the MSE and RMSE ordering with the single difference being the juxtaposition of RFR-ET and ABR-ET. The differences between the three indicators were not significant except for XGBR-ET, RFR-ET, and BR-ET. *R*^2^ for the eight models was ordered XGBR-ET > GBR-ET > SVR-ET > ABR-ET > BR-ET > LR-ET > KNR-ET > RFR-ET. 

*R*^2^ was greatest for XGBR-ET (0.981). *R*^2^ for XGBR-ET was 2.19% greater than for GBR-ET, which was second to it in prediction accuracy and 21.86% greater than for RFR-ET, which was least accurate. This analysis shows that the prediction accuracy of GBR-ET was satisfactory. XGBR-ET had the least MSE, RMSE, MAE, and MAPE and greatest *R*^2^ values, in contrast to other models, which indicates that the XGBR-ET model had the highest prediction accuracy.

### 2.4. Weather Factor Ablation Experiment

Ablation analysis was conducted to determine if the XGBR-ET model was overfitting. The importance of the factors was assessed and the importance of the rankings of the independent variables in the XGBR-ET model was analyzed (Figure 7a,b). Quantification of feature importance allowed us to obtain the degree of importance of the factors so that we could create permutations of the importance in order to weight and evaluate the importance of the features. Figure 7a shows that the features (independent variables) were ordered, in descending order of importance, *R_n_* > RH > RH_min_ > *T_amax_* > RH_max_ > *T_amin_* > *T_a_* > V, and after the permutation importance analysis of the test set (Figure 7b), it can be seen that the permutation importance of the eight independent variables for ET was *R_n_* > RH > RH_min_ > *T_amax_* > *T_amin_* > RH_max_ > Ta > V. It can thus be seen that in the XGBR-ET model, the independent variable *R_n_* had the greatest effect on model performance with an MDI value of 0.867, and V had least effect with an MDI value of 0.003.

The preceding analysis suggested that we conduct a series of ablation experiments based on the permutation importance order *R_n_* > RH > RH_min_ > *T_amax_* > *T_amin_* > RH_max_ > *T_a_* > V. The results of the training set and test set ablation experiments based on this order are shown in Figure 8. Num_Params 8 on the horizontal axis indicates that all eight independent variables were model inputs; Num_Params 7 indicates that model input excluded the least important permutation parameter, V: Num_Params 6 indicates that the two least important permutation parameters, *T_a_* and V, were excluded, and so on. The vertical axis shows MSE for the model predicted values versus the measured values.

Figure 8 shows that for the test set, MSE for the model output tended to decrease and then increase as the number of input variables was reduced, which indicates that the model over-fits (MSE = 0.057) when all variables were used as input terms (Num_Params 8 on the horizontal axis). As the number of input terms decreased, the model prediction accuracy reached the highest value (MSE = 0.047) at Num_Params 5 (i.e., the input variables were *R_n_*, RH, RH_min_, *T_amax_*, and *T_amin_*), and MSE increased if the number of input variables further decreased. The preceding ablation experiments led us to conclude that when modeling greenhouse drip irrigated tomato ET using XGBoost regression, *R_n_*, RH, RH_min_, *T_amax_*, and *T_amin_* should be selected as the model input variables to ensure the greatest model accuracy.

## 3. Discussion

Eight meteorological factors were used as variables (*R_n_*, *T_a_*, *T_amin_*, *T_amax_*, RH, RH_min_, RH_max_, and V) in this study. Analysis of the correlations between meteorological factors and ET showed that the meteorological factors affecting ET were primarily *R_n_*, RH, and Ta. This result is consistent with previous studies [31]. *R_n_* had the greatest effect on ET because net radiation, which is responsible for temperature differences, is the sole source of energy, and *R_n_* is related to sunshine hours and total solar radiation [32]. ET had a good fit with *T_amax_* and a bad fit with Ta and *T_amin_*; this result is consistent with the findings of Cheng et al. (2021) [33] and Wang et al. (2017) [34]. Huang and Li (2021) [35] used correlation analysis and principal component analysis to determine the contributions of meteorological factors to reference crop ET and found that ET0 had the greatest correlation with *T_amax_*, which is consistent with our results. We found that the meteorological factor V was least correlated with ET. This result differed slightly from that of Su and Fan (2020) [32] due to differences in the greenhouse environment and in experimental conditions.

In comparing and analyzing the results of fitting eight regression algorithms to ET, we found that among the eight models, MSE, RMSE, MAE, MAPE, and *R*^2^ for the linear models (LR-ET, SVR-ET, and KNR-ET) all indicated that SVR-ET had the best prediction accuracy. Although *R*^2^ for ABR-ET, BR-ET, and RFR-ET was <0.95, considering MSE, RMSE, MAE, MAPE, and *R*^2^ together showed high prediction accuracy for ABR-ET, BR-ET, and RFR-ET. Similar conclusions were made by Song et al. (2020) [18] and Li et al. (2019) [22]. *R*^2^ for GBR-ET reached 0.960, and prediction accuracy was also high, but *R*^2^ for XGBR-ET was greater, reaching 0.981, and was the highest of the eight models. We found that XGBR-ET was a powerful computational tool, and it continuously calculated the weights of hidden states from GBR, showing that it could better capture time series features and mine deeper information than conventional machine learning models. Additionally, XGBR-ET had the least values of MSE, RMSE, MAE, and MAPE, indicating that XGBR-ET more accurately predicted ET. XGBR-ET produced the best predictions because it increased the control of model complexity by randomly sampling samples and features, which improved the generalizability of the model and ultimately reduced the prediction error. Yu et al. (2020) [36] obtained results consistent with ours using XGBoost as the main regression model and a particle swarm optimization algorithm to optimize the parameters of XGBoost for meteorological and soil moisture data. In contrast, the ABR-ET and KNR-ET model architectures were relatively simple, and those models had limited ability to capture features. The prediction accuracy of the models, in descending order, was XGBR-ET > GBR-ET > SVR-ET > ABR-ET > BR-ET > LR-ET > KNR-ET > RFR-ET. In summary, XGBR-ET outperformed the other seven models in terms of accurate prediction of greenhouse tomato crop ET. We therefore recommend the XGBR-ET model for predicting ET for tomato crops and our results can be used to determine the best irrigation program for tomato crops in central China.

The importance of meteorological factors as independent variables in the XGBR-ET model was, in descending order, *R_n_* > RH > RH_min_ > *T_amax_* > RH_max_ > *T_amin_* > Ta > V. The importance value of *R_n_* was 0.867 and of RH was 0.092; the values of other factors were <0.035, which shows that *R_n_* and RH were more important than other factors of ET. Liu et al. (2020) [37] obtained similar results in a study of the effects of environmental factors on ET, but their conclusions differed slightly from those of Zhang et al. (2014) [38]. The main reason for this difference was that Zhang et al. (2014) [38] used the Penman–Monteith equation as the basic equation for calculating ET of greenhouse cucumber and used correlation analysis to calculate the amount of cucumber transpiration. Our results are slightly different in this study because the subject plant was tomato, and a machine learning algorithm was used. The ablation experiment showed that choosing *R_n_*, RH, RH_min_, *T_amax_*, and *T_amin_* as the model input variables ensured the greatest model accuracy.

## 4. Materials and Methods

### 4.1. Experimental Site Overview and Design

The experiment was conducted in a solar greenhouse at the comprehensive experimental base of Xinxiang, Chinese Academy of Agricultural Sciences (35°9′ N, 113°5′ E, 78.7 m above sea level), from March to July in each of the years 2019–2021. The soil in the test area was sandy loam (IUSS Working Group WRB, 2014) up to 1.0 m in depth, with a mean bulk density of 1.49 g/cm^3^, and field soil water capacity of 0.32 cm^3^/cm^3^. The tomato variety used in the trial was Jinpeng M6. Seedlings were raised in January and planted in the greenhouse in a full bed with an area 17.6 m^2^ (8.8 m long × 2 m wide) in wide–narrow rows (65 cm wide and 45 cm narrow) when they reached 3 leaves and 1 heart (3 complete leaves and 1 still developing cotyledon). The plants were irrigated by drip irrigation (drip head spacing 30 cm and drip head flow rate 1.1 L/h). No experimental treatment was set up at the seedling stage, and water treatment (irrigation quota was 0.9 *E_p_*) was started when tomatoes entered the rapid growth period and the soil moisture content at 0–60 cm dropped to 75% field capacity. The amount of water per irrigation event and irrigation frequency were determined by reference to cumulative evaporation (*E_p_*) of a standard 20 cm evaporation pan (20 cm diameter and 11 cm deep) that was placed 20 cm above the canopy, with height being adjusted promptly according to crop growth. At 07:30–08:00 everyday, evaporation from the pan was measured with an accuracy of 0.1 mm, and the pan was replenished after measurement from a quantity of 20 mm distilled water to ensure the water in the pan was free of impurities. When cumulative evaporation reached 20 ± 2 mm, the plot was uniformly irrigated. A water meter with accuracy of 0.001 m^3^ was installed at the head of the test site to ensure precise control of irrigation water. Other management measures were the same as conventional local greenhouse cultivation, as were all other agronomic measures, such as topping, spraying, and fruit counting.

### 4.2. Experimental Data Observation Content and Methods

#### 4.2.1. Meteorological Data

A fully automated meteorological monitoring system was installed in the middle of the greenhouse 2 m above the ground surface. The equipment included a net radiometer (*R_n_*, NRLITE2, Kipp & Zonen, Delft, The Netherlands), an integrated temperature–humidity sensor (*T_a_*, RH, CS215, Campbell Scientific, Inc., Monterrey, CA, USA), and an anemometer (V, Wind Sonic, Gill, UK) to measure wind speed at 2 m above the ground with an accuracy of ±0.02 m/s. All data were collected at 5 s intervals, and averages were calculated once every 30 min and recorded in the CR1000 data collector (Campbell, Monterrey, CA, USA).

#### 4.2.2. Soil Water Content

In order to monitor the changes of soil moisture and heat environment at different points in real time, a set of soil moisture monitoring systems (ZL6, Ningbo, China) were buried in the middle of each test plot, and each instrument was connected to 6 probes. The measurement depths were 0, 10, 20, 30, 40, and 60 cm, and the data were recorded every 30 min and stored in the system. In order to reduce the measurement error, water content of the 0–100 cm soil layer was measured using TRIME-IPH time domain reflectometry (Micromodultechnik GmbH, Ettlingen, Germany) in 20 cm layers; each measurement was repeated three times and the average value was taken. The TRIME tube was buried equidistant from two drip heads in the same drip irrigation belt and measured once before and once after irrigation. The instrument was corrected periodically at each reproductive stage by the soil drying method for instrument error reduction.

#### 4.2.3. Estimation of Tomato Evapotranspiration

Estimation of tomato diurnal evapotranspiration was calculated using the water balance method:(1)$$ET=P+I_{r}+U-D+(W_{0}-W_{t})$$
where *ET* is diurnal evapotranspiration (mm); *I_r_* is irrigation water (mm) (12 irrigations throughout the whole growth stage); *P* is rainfall (mm); *U* is groundwater recharge (mm); *D* is deep seepage (mm); and *W_0_* and *W_t_* are diurnal water storage in the 0–100 cm soil layer at the beginning and end of the period, respectively (mm). The water storage in the 0–100 cm soil depth was calculated by means of average volumetric water content. Since the experiment was conducted in a greenhouse, *P* = 0. The water table at the test site was deep (below 5.0 m), and groundwater could not be absorbed by the crop (i.e., *U* = 0). The single irrigation quota for all treatments was small (maximum 20 mm) and produced almost no deep seepage (i.e., *D* = 0). The above equation can therefore be simplified as:(2)$$ET=I_{r}+(W_{0}-W_{t})$$

### 4.3. The Eight Regression Algorithms

#### 4.3.1. XGBoost Regression

XGBoost was first proposed by Chen and Guestrin [39] as an improvement over the gradient boosted decision tree GBDT. Conventional trees only use first-order derivatives, but XGBoost regression (XBR) innovatively introduced second-order derivatives and regular terms, making the algorithm good in training and rapid in computing. The XBR learning process is as follows.
XGBoost basic function

Assume that there are *K* trees in the model. The basic function of the model can be expressed as:(3)$${\hat{y}}_{i}=\phi \left(x_{i}\right)=\overset{K}{\underset{k=1}{\sum }}f_{k}\left(x_{i}\right),f_{k}\in F\;with\;F=\left\{f\left(x\right)=\omega_{q\left(x\right)}\right\}\left(q:R^{m}\to \left\{1,2,\cdots ,T\right\},\omega \in R^{T}\right)$$
where ${\hat{y}}_{i}$ is the prediction value of the model for the sample, *F* denotes the set of all trees, *f*(*x*) is the function of one tree, *T* is the number of leaf nodes of the tree, *q(x)* is the mapping function of the sample data corresponding to a leaf node on the tree, and *w_q_*_(*x*)_ is the score of the leaf node.


2.XGBoost objective function


(4)$$Obj(\theta )=\sum_{i}^{n}l\left(y_{i},{\hat{y}}_{i}\right)+\sum_{k=1}^{K}\Omega \left(f_{k}\right)$$(5)$$\Omega \left(f_{k}\right)=\gamma T+\frac{1}{2}\lambda \overset{T}{\underset{j=1}{\sum }}w_{j}^{2}$$
where $\overset{n}{\underset{i}{\sum }}l\left(y_{i},{\hat{y}}_{i}\right)$ is the model loss function, $\Omega \left(f_{k}\right)$ is the regular term of tree *k*, and *γ* and *λ* are XGBoost customizing parameters that, respectively, limit the number of leaf nodes and control the size of the node score; other variables are as for the previous equations.


3.XGBoost training


The XGBoost algorithm is an ensemble technique that trains cumulatively and successively to iteratively optimize the objective function until the objective function reaches a minimum value, at which time training is complete. The training process starts with the optimization of the first tree, and when the model iterates to tree *t*, it is given by:(6)$${\hat{y}}_{i}^{\left(t\right)}=\overset{t}{\underset{k=1}{\sum }}f_{k}\left(x_{i}\right)={\hat{y}}_{i}^{\left(t-1\right)}+f_{t}\left(x_{i}\right)$$

If the loss function is squared error, the objective function can be changed to:(7)$$Obj^{\left(t\right)}=\sum_{i}^{n}{\left(y_{i}-{\hat{y}}_{i}^{\left(t\right)}\right)}^{2}+\overset{t}{\underset{i=1}{\sum }}\Omega \left(f_{t}\right)\;=\sum_{i}^{n}\left[2\left({\hat{y}}_{i}^{\left(t-1\right)}-y_{i}\right)f_{t}\left(x_{i}\right)+f_{t}{\left(x_{i}\right)}^{2}\right]+\Omega \left(f_{t}\right)+constant$$
where ${\hat{y}}_{i}^{\left(t\right)}$ is the prediction value of the model that has iterated to *t* trees, ${\hat{y}}_{i}^{\left(t-1\right)}$ is the prediction of the model after optimization of the previous *t* − 1 trees, $f_{t}\left(x_{i}\right)$ is the score of the newly added *t* trees, and *constant* is the sum of the regularization terms of the previous *t* − 1 trees.

If the loss function is a general function, the Taylor expansion is used to solve for the minimum value.

#### 4.3.2. Linear Regression

A multivariate linear regression model is expressed in the form:(8)$$Y_{i}=\beta_{0}+\beta_{1}X_{i1}+\beta_{2}X_{i2}+\cdots +\beta_{p}X_{ip}+\varepsilon_{i}\;\;\;\;\;\;\;\;\;\;i=1,2,\cdots ,n.$$
where $X_{i1}$, $X_{i2}$, …, $X_{ip}$ are independent predictor variables and $Y_{i}$ is the predicted outcome (dependent) variable.

#### 4.3.3. Support Vector Regression

The support vector regression model (SVR) [40] is:(9)$$f\left(x\right)=\overset{n}{\underset{i=1}{\sum }}\left(a_{i}-a_{i}^{*}\right)k(x_{i}\cdot x)+b$$
where $a^{*}$ is the Lagrangian multiplier; *i* is sample *i*, and $x_{i}\in R^{N}$.

#### 4.3.4. K Neighbors Regression

K neighbors regression (KNR) is used to find the closest *K* samples in the training set based on a distance measure for each sample pair, and then to make predictions based on the *K* nearest neighbors.

#### 4.3.5. Random Forest Regression

Random forest regression (RFR) is a combinatorial classification method that is based on decision trees and statistical learning theory [41]. *M* samples are randomly selected by bagging (bootstrap aggregating), and *m* variables are then randomly selected at each node as candidates for splitting the node to construct a single decision tree. These steps are repeated to generate a mass regression decision tree. The final prediction result of the model is the average of the prediction results of the mass regression decision tree.

#### 4.3.6. AdaBoost Regression

AdaBoost is an iterative augmented regression algorithm (ABR) [42]. It uses a single feature as a weak learning algorithm and performs multiple iterations on the same training sample set to form a sequence of weak classifiers, then selects weights based on the classification effect and finally weights and combines the classifiers to form a strong classifier.
(10)$$H\left(x\right)=\left\{\begin{array}{c} 1\;\;\;\;\;\;\;\;\;\;\;\;\;\;\;\;\;\;\;\;\;\;\;\sum_{t=1}^{T}\alpha_{t}h_{t}\left(x\right)\geq 0.5\sum_{t=1}^{T}\alpha_{t}\\ 0\;\;\;\;\;\;\;\;\;\;\;\;\;\;\;\;\;\;\;\;\;\;\;\;\;\;\;\;\;\;other\;\;\;\;\;\;\;\;\;\;\;\;\;\;\;\;\;\;\;\;\;\;\;\;\;\;\;\;\;\;\;\;\;\; \end{array}\right.$$
where *T* is the number of iterations, and $\alpha_{t}=ln(\frac{1}{\beta_{t}})$ is the weight of the weak classifier $h_{t}$ obtained in iteration *t*. A larger value of *H*(*x*) indicates greater importance of the weak classifier.

#### 4.3.7. Bagging Regression

Bagging regression (BR) [43] is based on the random selection of *k* training sets from a dataset to obtain weak classifiers and then training and voting to obtain strong classifiers for subsequent processing.

#### 4.3.8. Gradient Boosted Regression

The integrated gradient boosting regression (GBR) algorithm is, after *M* iterations [20]:(11)$$f_{M}\left(x\right)=\rho +0.1\overset{M}{\underset{m=1}{\sum }}\overset{J}{\underset{j=1}{\sum }}\rho_{mj}I,\;\;\;\;\;\;\;\;x\in R_{mj}$$
where $R_{mj}$ is the leaf node region, *j* = 1, 2, …, *j*; $\rho_{mj}$ is the best residual fit value; and *I* is the indicator function, which takes the value 1 when condition *x* falls into the leaf node region and 0 otherwise.

### 4.4. Data Processing and Model Evaluation

The data for 2019–2021 were combined and randomized and then divided into two datasets for training and testing in the ratio 4:1. The XGBoost regression algorithm was called using Python for model training and testing.The data were standardized before modeling in order to eliminate the influence of the magnitude between indicators on the prediction using the equation:(12)$$Y=\frac{X-\bar{X}}{\sigma }$$
where Y is the standardized value, *X* is the original data, $\bar{X}$ is the mean of the original data, and σ is the variance of the original data.

3.Mean square error (MSE), root mean square error (RMSE), mean absolute error (MAE), mean absolute percentage error (MAPE) [44,45,46], and coefficient of determination (*R*^2^) were used to evaluate the accuracy of the model [47]. Lower values of MSE, RMSE, MAE, and MAPE indicate greater prediction accuracy. *R*^2^ indicates the degree of fit between predicted and measured values of the model; if *R*^2^ is close to 1, the model is a good fit. The equations are:(13)$$MSE=\frac{\sum_{1}^{N}{\left(Q_{i}-P_{i}\right)}^{2}}{N}$$(14)$$RMSE={\left[\overset{N}{\underset{i=1}{\sum }}\frac{{\left(Q_{i}-P_{i}\right)}^{2}}{N}\right]}^{0.5}$$(15)$$MAE=\frac{1}{N}\overset{N}{\underset{i=1}{\sum }}\left|Q_{i}-P_{i}\right|$$(16)$$MAPE=\frac{100\%}{N}\overset{N}{\underset{i=1}{\sum }}\left|\frac{Q_{i}-P_{i}}{P_{i}}\right|$$(17)$$R^{2}=\left\{\frac{\sum_{i=1}^{N}\left(Q_{i}-\bar{Q}\right)\left(P_{i}-\bar{P}\right)}{\sqrt{(Q_{i}-{\bar{Q)}}^{2}}\sqrt{{\left(P_{i}-\bar{P}\right)}^{2}}}\right\}$$
where $P_{i}$ is the measured value, $Q_{i}$ is the model predicted value, $\bar{P}$ is the mean of the measured value, $\bar{Q}$ is the mean of the model predicted value, and *N* is the number of data points.

## 5. Conclusions

We measured daily ET, *R_n_*, *T_a_*, *T_amin_*, *T_amax_*, RH, RH_min_, RH_max_, and V of greenhouse tomatoes in order to analyze the meteorological factors that affected ET and to compare the accuracy of various models in predicting ET using eight regression algorithms: LR, SVR, KNR, RFR, ABR, BR, XGBR, and GBR. From analysis of our results, we drew the following conclusions.
*R_n_*, Ta, and *T_amax_* were positively correlated with ET, and *T_amin_*, RH, RH_min_, RH_max_, and V were negatively correlated with ET. *R_n_* had the greatest correlation with ET (r = 0.89), and V had the least correlation with ET (r = 0.43).Prediction accuracy of the models was, in descending order, XGBR-ET > GBR-ET > SVR-ET > ABR-ET > BR-ET > LR-ET > KNR-ET > RFR-ET. The respective values of MSE, RMSE, MAE, MAPE, and *R*^2^ for XGBR-ET were 0.032, 0.163, 0.132, 4.47%, and 0.981. XGBR-ET was more accurate in predicting ET than the other seven models. Thus, the XGBR-ET model better predicts daily evapotranspiration of the greenhouse tomato crop during the entire growth period.The results of the ablation experiments showed that the feature importance of the input variables of XGBR-ET was, in descending order, *R_n_* > RH > RH_min_ > *T_amax_* > RH_max_ > *T_amin_* > *T_a_* > V. When predicting ET of drip irrigated greenhouse tomato using XGBR, the selection of *R_n_*, RH, RH_min_, *T_amax_*, and *T_amin_* as model input variables will ensure maximum accuracy (MSE = 0.047).

## Figures and Tables

**Figure 1 plants-11-01923-f001:**
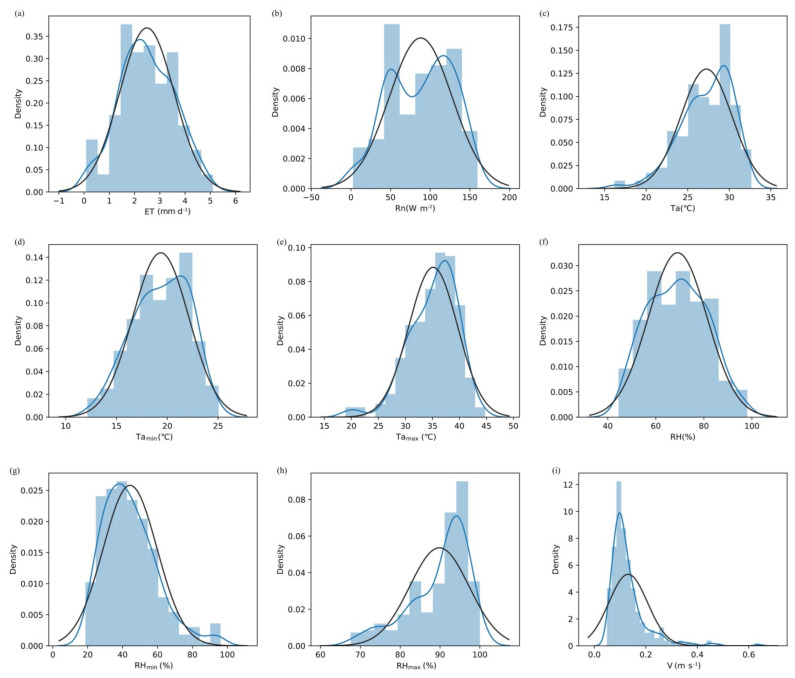
Distributions of ET and eight meteorological factors. (**a**) showing the distribution of evapotranspiration (ET). (**b**) showing the distribution of net solar radiation (*R_n_*). (**c**) showing the distribution of temperature (*T_a_*). (**d**) showing the distribution of minimum temperature (*T_amin_*). (**e**) showing the distribution of maximum temperature (*T_amax_*). (**f**) showing the distribution of relative humidity (RH). (**g**) showing the distribution of minimum relative humidity (RH_min_). (**h**) showing the distribution of maximum relative humidity (RH_max_). (**i**) showing the distribution of wind speed (V). The blue curve showing the fitting of the normal distribution for each factor. The black curve is the fitted standard distribution curve.

**Figure 2 plants-11-01923-f002:**
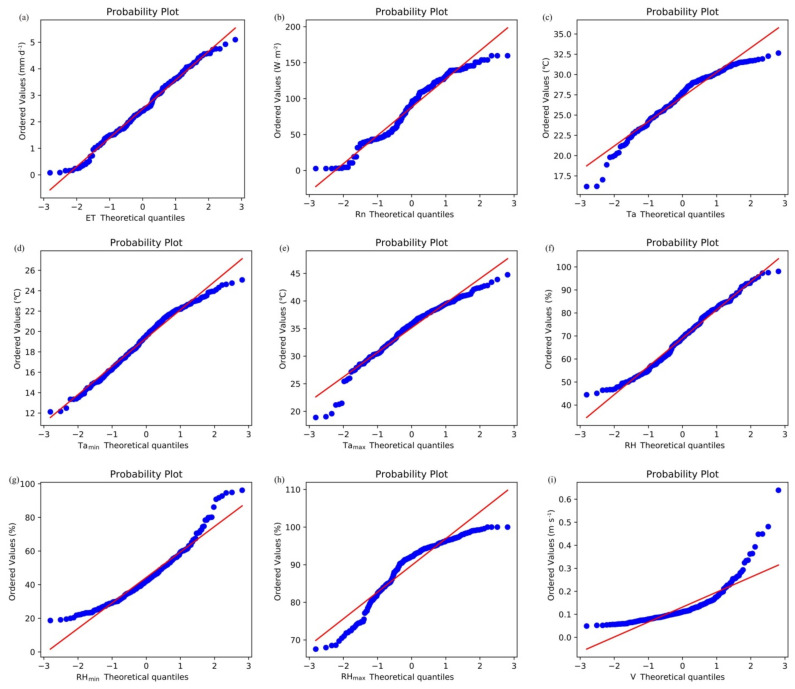
Normal probability plot of ET and eight meteorological factors. (**a**) showing the normal probability plot of evapotranspiration (ET). (**b**) showing the normal probability plot of net solar radiation (*R_n_*). (**c**) showing the normal probability plot of temperature (*T_a_*). (**d**) showing the normal probability plot of minimum temperature (*T_amin_*). (**e**) showing the normal probability plot of maximum temperature (*T_amax_*). (**f**) showing the normal probability plot of relative humidity (RH). (**g**) showing the normal probability plot of minimum relative humidity (RH_min_). (**h**) showing the normal probability plot of maximum relative humidity (RH_max_). (**i**) showing the normal probability plot of wind speed (V).

**Figure 3 plants-11-01923-f003:**
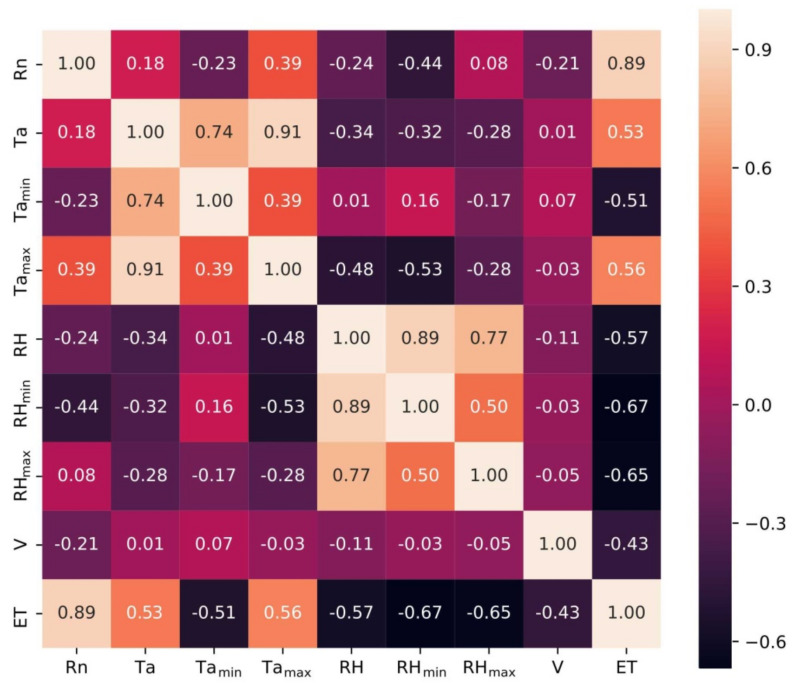
Correlation coefficients for ET and eight meteorological factors.

**Figure 4 plants-11-01923-f004:**
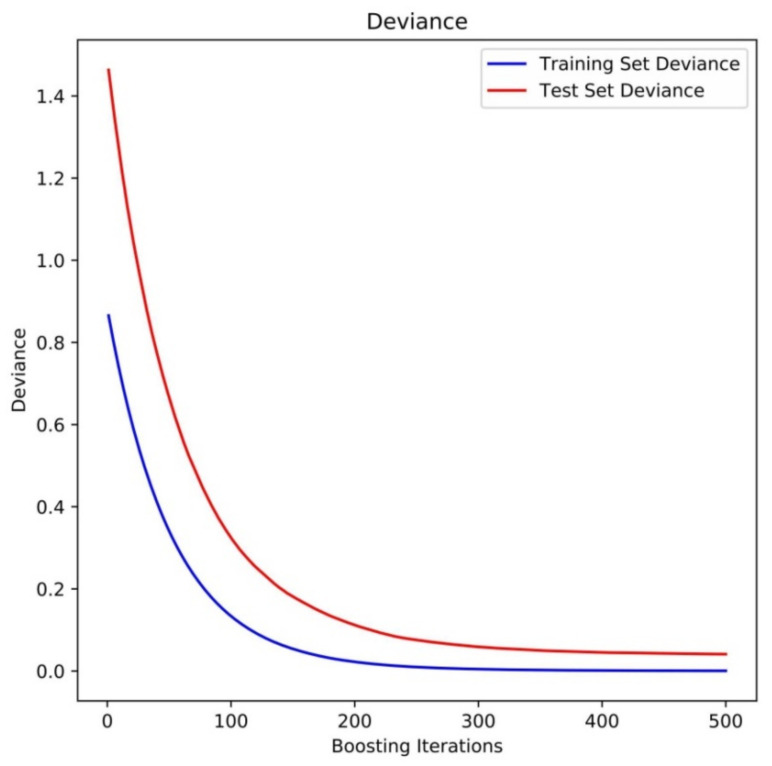
XGBR training loss.

**Figure 5 plants-11-01923-f005:**
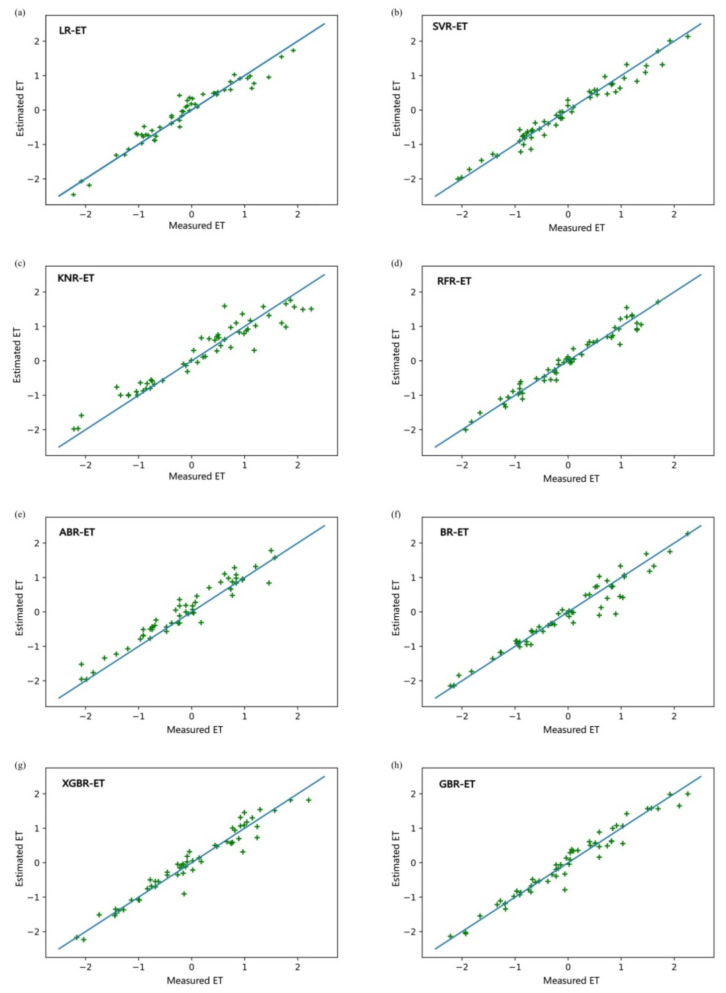
Fitting results of ET predicted by eight models. (**a**) fitting result of measured ET and estimated ET by LR-ET. (**b**) fitting result of measured ET and estimated ET by SVR-ET. (**c**) fitting result of measured ET and estimated ET by KNR-ET. (**d**) fitting result of measured ET and estimated ET by RFR-ET. (**e**) fitting result of measured ET and estimated ET by ABR-ET. (**f**) fitting result of measured ET and estimated ET by BR-ET. (**g**) fitting result of measured ET and estimated ET by XGBR-ET. (**h**) fitting result of ET measured ET and estimated ET by GBR-ET.

**Figure 6 plants-11-01923-f006:**
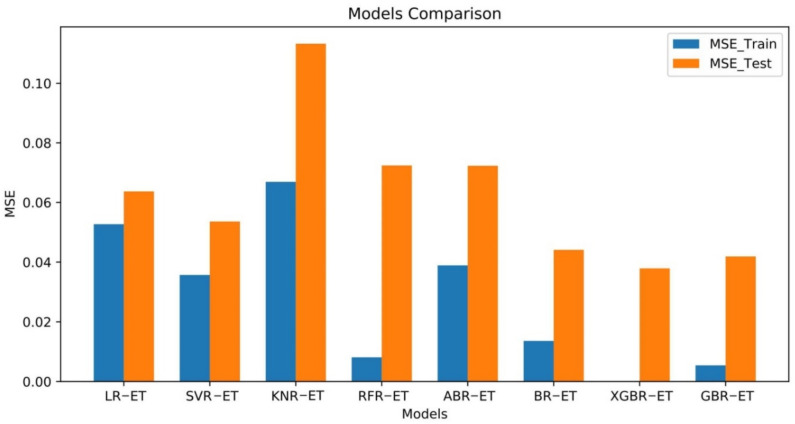
MSE values for ET predicted by training and testing datasets for eight models.

**Figure 7 plants-11-01923-f007:**
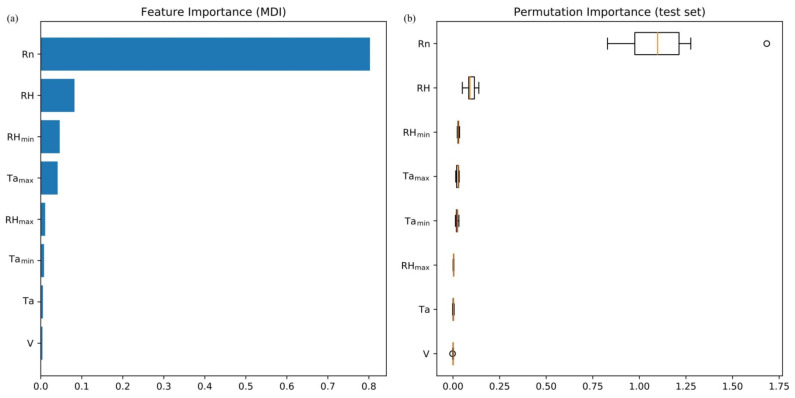
Plots of the characteristic importance and ranking importance of the input variables of the XGBR-ET model. (**a**) Plot of the characteristic importance of the input variables of the XGBR-ET model. (**b**) Plot of the ranking importance of the input variables of the XGBR-ET model.

**Figure 8 plants-11-01923-f008:**
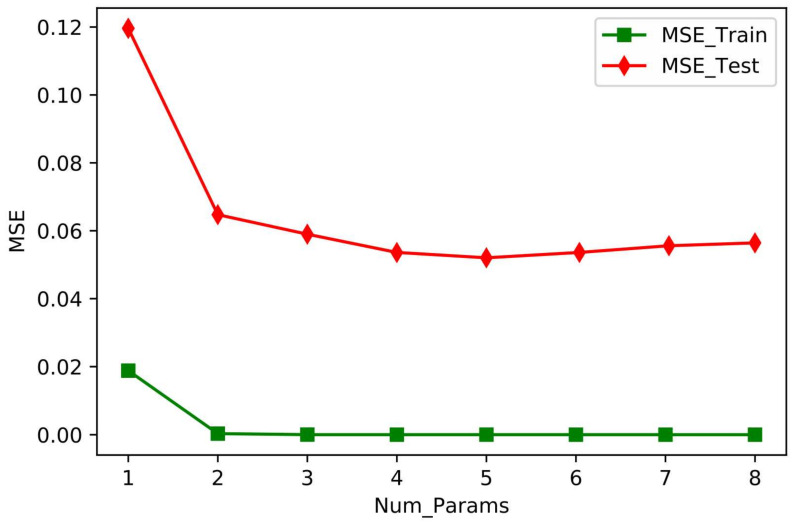
MSE plots of eight meteorological factors in ablation experiments.

**Table 1 plants-11-01923-t001:** Values of MSE, RMSE, MAE, MAPE, and *R*^2^ for the eight models.

Model	MSE	RMSE	MAE	MAPE	*R* ^2^
LR-ET	0.067	0.257	0.172	8.36%	0.812
SVR-ET	0.053	0.218	0.162	6.72%	0.854
KNR-ET	0.115	0.303	0.237	10.83%	0.807
RFR-ET	0.072	0.285	0.197	8.77%	0.805
ABR-ET	0.071	0.282	0.216	9.95%	0.834
BR-ET	0.042	0.207	0.159	6.59%	0.823
XGBR-ET	0.032	0.163	0.132	4.47%	0.981
GBR-ET	0.039	0.205	0.154	6.38%	0.960

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
