# Peer review of "Prediction of Greenhouse Tomato Crop Evapotranspiration Using XGBoost Machine Learning Model"

_plants, 2022, doi:10.3390/plants11151923_

Round 1

Reviewer 1 Report

REVIEWER’S REPORT

Manuscript number: MPDI plants-1814461

Title: Prediction of greenhouse tomato crop evapotranspiration using XGBoost machine learning model

General comments:

Thank you very much for the opportunity to review this paper. The paper is very interesting as not much studies were being done on improving crop water use efficiency for drip-irrigated crops grown under a greenhouse condition. The determination of accurate ET is vital deciding the timing and amount of irrigation to attain high water use efficiency and water productivity. Using machine learning models, the paper aimed at developing a crop ET prediction model (XGBR-ET) based on XGBoost Regression and compared the model prediction performance with seven other regression models. Three years of experimental data with eight meteorological parameters (including ET) were used to train and test the model.  These meteorological parameters were standardized, checked if they were normally distributed, and correlations between these variables were performed. The results indicated that XGBR-ET produced the highest prediction accuracy using statistical indicators. Model input variables Rn, RH , RHmin, Tamax and Tamin were enough to produce accurate XGBR-ET model prediction.

The paper was written with sufficient number of up-to-date references to back up the results of the studies. I am pleased that this valuable paper is being submitted for publication consideration in MDPI.  However, I have some comments and questions on the paper for the authors to consider and respond. Given the significance of the paper, I would like to recommend its acceptance for publication, however, the authors should address my specific comments.

Specific COMMENTS:

Line 9-10:  Revise this sentence into a stronger rationale.

Line 100-106: I suggest to re-write the objectives and make them “smart” objectives.

Line 118-120: What were the water treatments mentioned here? Briefly Describe them here.

Line 131: In the experimental data used in the analysis and models, crop growth stage was not included in the analysis considering that actual crop ET is strongly influenced by crop growth stages. If there is no consideration of crop growth stages, then the model may just predict the reference or potential ET, not the actual crop ET. Actual ET is the product of reference ET and crop coefficient, where crop coefficient varies with growth stages. Please justify the exclusion of crop growth stages.

Line 147: Is the evapotranspiration an actual crop evapotranspiration? If yes, make a statement that ET means the actual ET, and whenever ET refers to as reference ET, mention it also.

Line 157: For the ET equation, briefly describe how change in storage (as an absolute value of the change in soil water storage was computed/determined? This should be from the soil water content data.

Reviewer 2 Report

Dear Authors,

I revised the manuscript "Prediction of greenhouse tomato crop evapotranspiration using XGBoost machine learning model" submitted to Plants journal. The manuscript is interesting, with a well-conducted discussion of the results. However, I have some concerns which need to be addressed before considering for final publication.

General comments:

1. References should be numbered in order of appearance and indicated by a numeral or numerals in square brackets—e.g., [1] or [2,3], or [4–6].

2. The order of sections should be as follows: 1. Introduction, 2. Results, 3. Discussion, 4. Materials and Methods, 5. Conclusions.

Minor comments:

1. Line 1. Line 1. Use only one type of paper - Article.

2. Section "1. Introduction". The market role of the tomato should be more broadly described in this section. In addition, there is a lack of citation of tomato crop models.

3. Equation descriptions need to be completed, e.g. line 198, equation 8 - the description for "??" is missing.

4. Section "2. Materials and Methods". It is worth adding another performance index of error prediction, which is MAPE (Mean Absolute Percentage Error), because it is expressed as a percentage. In agricultural research, ranges of MAPE indicator values can be found that show the practical usefulness of the model. For convenience, refer to the following papers: https://doi.org/10.3390/land10060609, https://doi.org/10.1016/j.agrformet.2015.03.007, https://doi.org/10.3390/su12051763.

5. Figures 1, 2, 3, 4, 7 and 8 should be moved below the text in which they are referenced.

6. Section "Author Contributions" is missing.

7. Section "References". The style of the references is not in accordance with the requirements of Plants. I suggest using a bibliography software package like Mendeley, Zotero, EndNote etc. Check out the instructions for authors at https://www.mdpi.com/files/word-templates/plants-template.dot.
